# Learning from electronic prescribing errors: a mixed methods study of junior doctors' perceptions of training and individualised feedback data

Ann Chu ,[1] Arika Kumar,[2,3] Geraldine Depoorter,[2] Bryony Dean Franklin,[2,3] Monsey McLeod [2]

¹Faculty Education Office, Imperial College London, London, UK
²Department of Pharmacy, Imperial College Healthcare NHS Trust, London, UK
³UCL School of Pharmacy, London, UK

**Correspondence to**
Dr Ann Chu;
annchu@doctors.org.uk

## ABSTRACT

**Objectives** To explore the views of junior doctors towards (1) electronic prescribing (EP) training and feedback, (2) readiness for receiving individualised feedback data about EP errors and (3) preferences for receiving and learning from EP feedback.

**Design** Explanatory sequential mixed methods study comprising quantitative survey (phase 1), followed by interviews and focus group discussions (phase 2).

**Setting** Three acute hospitals of a large English National Health Service organisation.

**Participants** 25 of 89 foundation year 1 and 2 doctors completed the phase 1 survey; 5 participated in semi-structured interviews and 7 in a focus group in phase 2.

**Results** Foundation doctors in this mixed methods study reported that current feedback provision on EP errors was lacking or informal, and that existing EP training and resources were underused. They believed feedback about prescribing errors to be important and were keen to receive real-time, individualised EP feedback data. Feedback needed to be in manageable amounts, motivational and clearly signposting how to learn or improve. Participants wanted feedback and better training on the EP system to prevent repeating errors. In addition to individualised EP error data, they were positive about learning from general prescribing errors and aggregated EP data. However, there was a lack of consensus about how best to learn from statistical data. Potential limitations identified by participants included concern about how the data would be collected and whether it would be truly reflective of their performance.

**Conclusions** Junior doctors would value feedback on their prescribing, and are keen to learn from EP errors, develop their clinical prescribing skills and use the EP interface effectively. We identified preferences for EP technology to enable provision of real-time data in combination with feedback to support learning and potentially reduce prescribing errors.

## BACKGROUND

Prescribing errors are reported to affect 7% of inpatient medication orders.[1] Worldwide, medication errors comprise a global patient safety issue, costing an estimated £33 billion.[2]

### STRENGTHS AND LIMITATIONS OF THIS STUDY

⇒ A key strength is the sequential explanatory mixed methods approach that allowed further exploration of survey responses with in-depth qualitative data from interviews and focus groups.
⇒ All foundation doctors in the organisation were invited to participate; this enabled capture of a range of views and experiences of electronic prescribing (EP) training and feedback that could be used to identify targets for improvement.
⇒ Findings from a single organisation may not be generalisable to other hospital settings, but may be useful for those using similar EP systems and providing similar types of training.

Electronic prescribing (EP) is generally advocated for reducing prescribing errors but studies have typically focused on the technology and not the human factors or educational aspects.[3] A systematic review highlighted that EP systems can introduce other error types, including incomplete computerised displays, selection errors in drop-down menus, reliance on default settings and over-dependence on clinical decision support.[4] Therefore, the transition to EP in the English National Health Service (NHS) introduces an additional component for which staff require training.

Medication safety research based on theories of human error suggests that prescribing errors occur as a result of latent and environmental conditions aligning with, or contributing to, failures in defences.[5] Learning from prescribing errors requires an understanding of the context (both at system and individual level) and that feedback be provided for relevant stakeholders, including prescribers. Past research has generally focused on measuring prescribing errors or root cause analysis[6–10]; more recent publications on feedback

interventions to improve prescribing are still primarily in the context of paper drug charts.[11–14]

In the UK, newly qualified ('junior') doctors have previously been identified as having one of the highest error rates among prescribers, as well as doing most inpatient prescribing.[6 15] The UK Foundation Programme is a 2-year mandatory training programme for newly qualified medical graduates. The Royal College of Physicians recognises that supporting junior doctors in safe prescribing requires education, practical resources and a culture of patient safety.[16] Numerous studies have identified this novice prescriber group as a focus for educational interventions but to date, most have been in the context of paper-based prescribing.[3 11–14]

A previous questionnaire study of junior doctors and hospital pharmacists identified that receiving structured feedback, including prescribing rates, was well received.[17] However, this work was conducted in the context of paper-based prescribing and required resource-intensive manual audits; this resulted in considerable delay for feedback and was not sustainable for routine implementation. We have therefore identified a literature gap in relation to EP feedback for junior doctors. This study aimed to explore the views of junior doctors towards (1) the EP training and feedback they receive, (2) readiness for receiving individualised EP feedback data and (3) preferences for how they would like to receive and learn from EP feedback.

## METHODS
### Setting
We conducted a mixed methods study at three large hospitals within an English NHS organisation (trust). The organisation has approximately 10 000 staff serving a population of one million people. EP has been active in most inpatient areas since 2016. This study focused on one organisation so that participants were referring to the same EP system and had equal access to training resources.

Classroom-based training was offered at trust induction for junior doctors. Optional EP training resources include online learning modules (generic and specialty-specific), 'Quick Reference Guides' on the trust's intranet, some postgraduate teaching sessions and EP technical support. EP 'order sentences' with default doses were available for many commonly used medications; clinical decision support tools such as allergy and drug interaction checking were not in routine use at the time of the study.

### Participants
All foundation doctors in their first 2 years of practice (Foundation Years 1 (FY1) and 2 (FY2)) at the trust (n=89) were invited to participate. Participant invitations were circulated by the Trust postgraduate education team and blinded to the research team. The year groups were approximately equal in size.

### Study design
This was an explanatory sequential mixed methods study comprising two phases: phase 1 was a quantitative questionnaire delivered to all foundation doctors in February/March 2018 and phase 2 was a qualitative interview and focus group study with volunteer participants from the same population in April/May 2018. This study design was chosen to permit analysis of the quantitative data for broad perspectives followed by in-depth exploration of themes that required further explanation.

### Patient and public involvement
There was no patient and public involvement in this study.

### Data collection
Phase 1 questionnaires were distributed online and on paper. Email circulation via postgraduate education teams ensured that all foundation doctors working in the organisation had an opportunity to participate. The email summarised the purpose of the study and included a weblink to an online version of the questionnaire. In addition, paper questionnaires were distributed at teaching sessions.

In phase 2, the same population of foundation doctors were invited to participate in interviews. In addition, a lunchtime focus group was offered to maximise recruitment. Due to the anonymous nature of the survey, we did not confirm if any of the participants contributed to both phases. However, we decided that maximising recruitment for phase 2 was essential and invited the whole population to participate.

#### Phase 1—questionnaire study
The questionnaire comprised 65 questions (online supplemental appendix 1). This was based on a questionnaire used in a previous study.[17] There were two major adaptations: (1) questions were adapted to reference EP (rather than paper-based prescribing), and (2) clinical scenarios were added. Most responses required were on 5-point Likert scales ranging from 'strongly disagree' to 'strongly agree'. Where relevant, there were additional options for 'not used' or 'not available'. The questionnaire was not psychometrically tested.

Four clinical scenarios were devised by research pharmacists as examples of prescribing errors with different levels of harm. These were based on categories from a national incident reporting system comprising 'no harm', 'low harm', 'moderate harm' to 'severe harm'.[18] The fifth category of 'death' was not included. Clinical details were devised from clinical experience and knowledge of local prescribing governance committees. The scenarios were prepiloted by four pharmacists and one doctor for face and content validity.

#### Phase 2—qualitative interview and focus group study
Interviews and a focus group were conducted using a prepiloted topic guide (online supplemental appendix 2). This was formulated after an interim analysis of survey results in phase 1 and identified areas of further

interest. Six example prescribing error feedback formats were obtained from the published literature and shown to participants. A table of EP error types (eg, allergies, dosing errors) was also used as a prompt for discussion. Handwritten notes were recorded for both the focus group and the interviews, which included some verbatim comments from participants.

### Data analysis

Phase 1 quantitative data were summarised using descriptive statistics, using Microsoft Excel. Positive responses ('agree and strongly agree') and negative responses ('disagree and strongly disagree') were combined when reporting the data. Neutral responses were included in the figures.

Phase 2 interview and focus group notes were transcribed and then analysed thematically using an inductive approach. This involved systematic coding of the transcripts, organising codes into categories followed by grouping into broader themes (Appendix 3). Two researchers conducted this independently, with quality assurance by a third, more experienced researcher. Comparison between the interview and focus group data sets suggested that themes were similar and could be integrated. Themes were discussed to consensus. Member checking was not used.

The quantitative and qualitative data were then further integrated at the discussion phase. This included reflection on whether the data was complimentary or if there were areas of divergence. This was discussed to consensus by the wider research team.[19 20]

## RESULTS

In the phase 1 quantitative study, 25 of 89 (28%) foundation doctors submitted responses. Respondents comprised 14 women (56%), 8 men (32%) and 3 whom did not disclose their gender. The majority were FY1 (n=16), six were FY2 (n=6) and three did not indicate their grade. Respondents were from 14 different medical schools.

In the phase 2 qualitative study, 12 participants took part in either a focus group (n=7) or an individual interview (n=5). Eight were FY1 and four were FY2.

### Phase 1: quantitative survey data

#### Current level of EP training and feedback

Foundation doctors reported varied experiences of EP and associated training resources (figure 1). They were equivocal as to whether EP reduced the number of prescribing errors that led to patient harm (12/25 agreed). Similarly, about half of them were satisfied with the training that had been provided at induction (13/25). More than half of them were satisfied with the availability of resources to enable them to prescribe safely (18/25). However, there was a lack of awareness and/or use of specifically tailored training resources. Some reported not having received EP classroom training (7/25) and 13/25 reported not using the specialty-specific training modules. The majority reported not receiving the training provided as part of the postgraduate education sessions (14/25) or not having used the 'Quick Reference Guides' and crib sheets available on the intranet (17/25). However, in free-text responses, respondents perceived that training on the EP interface

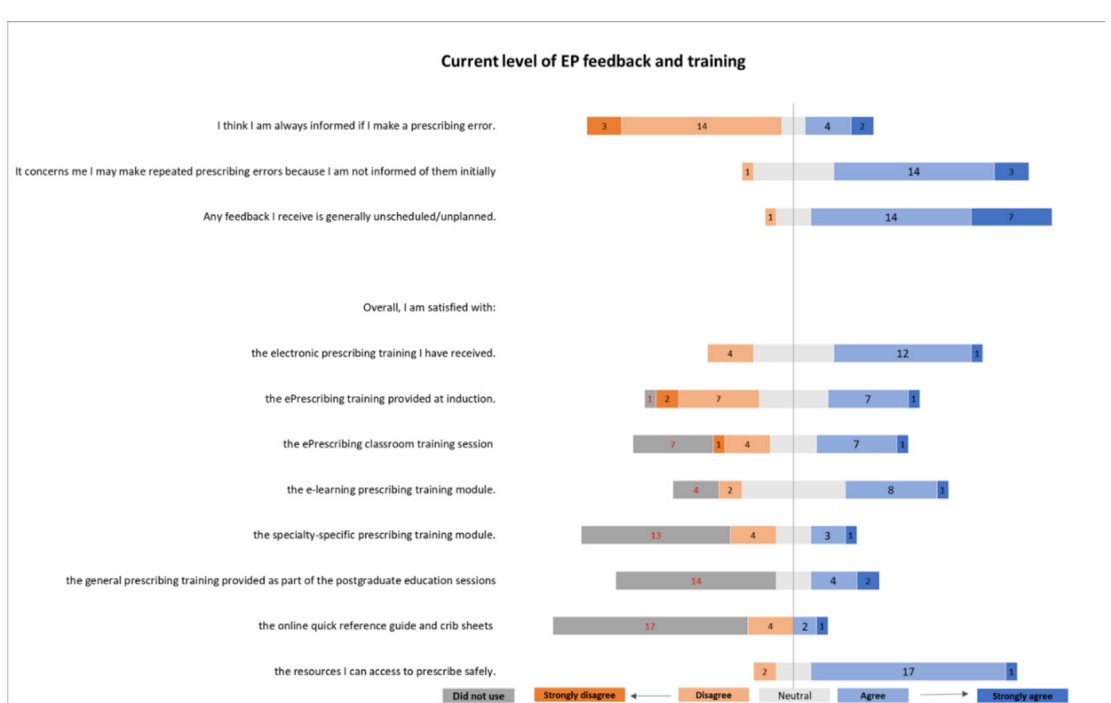

**Figure 1** Survey results: current level of electronic prescribing (EP) feedback and training reported by Foundation doctors (numbers correspond with number of respondents, total 25 participants).

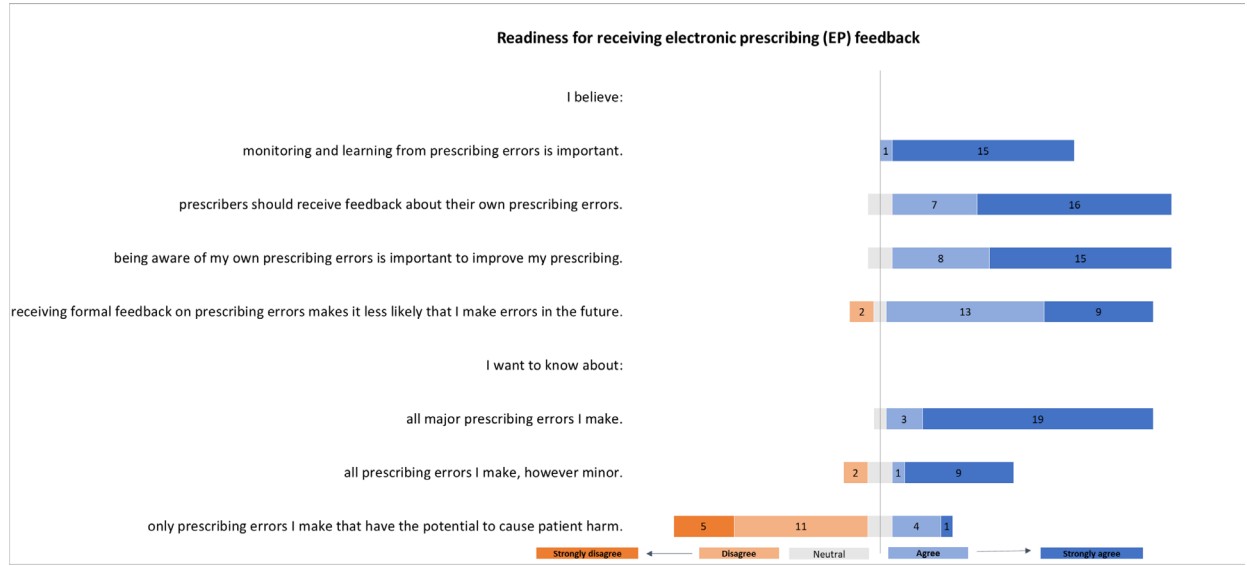

**Figure 2** Survey results: readiness for receiving EP feedback reported by Foundation doctors (numbers correspond with number of respondents, total 25 participants).

would be useful, and that better understanding of the EP system might help prevent errors.

The majority of the respondents said they were not always informed of their own prescribing errors (17/25) and that this was of concern. If they did receive feedback, this was generally in an unplanned or unscheduled way (21/25). Free-text responses reconfirmed that 'little' or 'rare' feedback was received. Junior doctors stated they would seek EP help from their peers (23/25), pharmacists (22/25) or more senior doctors (12/25). However, they said they generally did not ask nurses (6/25) or consultants (2/25).

### Readiness for receiving individualised EP feedback data

Most respondents (23/25) believed receiving feedback about prescribing errors was important and that being aware of their own errors would help improve their prescribing (figure 2). They believed receiving formal feedback on prescribing errors would reduce future errors (22/25). Foundation doctors wanted to know about all prescribing errors they made (22/25), including minor errors (19/25) and confirmed this when presented with example clinical scenarios covering 'no harm' (18/25) and 'low harm' (22/25).

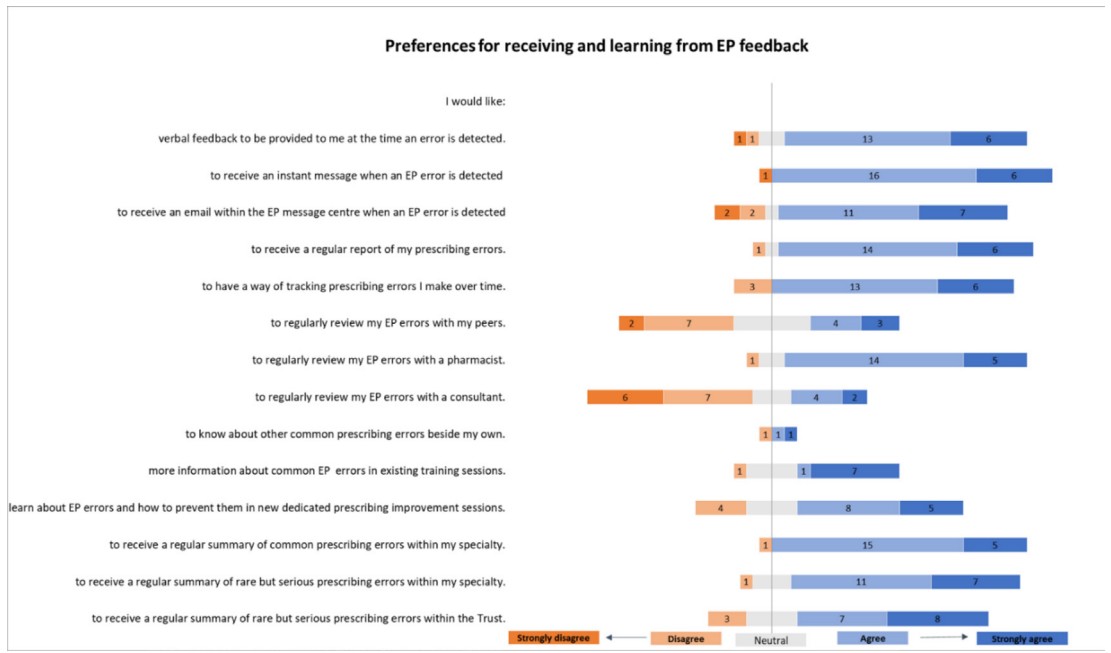

**Figure 3** Survey results: preferences for receiving and learning from electronic prescribing (EP) feedback reported by Foundation doctors (numbers correspond with number of respondents, total 25 participants).

## Preferences for receiving and learning from individualised EP feedback

Participants were keen for individualised feedback data to be timely; they supported a variety of methods including verbal (19/25), instant messaging (22/25) and/or EP system emails (18/25) (figure 3).

Respondents were receptive to the idea of receiving a regular report (20/25) and the ability to track errors made over time (19/25). They were keen to review personal EP errors with a pharmacist (19/25). However, there was a lack of consensus about doing this with their peers (7/25 agreed and 9/25 disagreed) and opposition to reviewing EP errors with a consultant (6/25 agreed and 13/25 disagreed).

Participants were also keen to learn from others' prescribing errors that were relevant to their practice (20/25). This information could be received in several ways, including on the ward (14/25), in existing training sessions (17/25) or new dedicated sessions (13/25). They were also receptive to receiving a regular summary of common prescribing errors within their specialty (20/25), rare but serious errors in their specialty (18/25), and in the organisation (15/25).

### Phase 2: qualitative interview and focus group data
#### Preferences for receiving and learning from EP feedback
Participants discussed both general prescribing feedback and their perceptions of EP-specific feedback. Participants' preferences for receiving and learning from prescribing feedback were grouped into four themes: (1) nature of individualised feedback data, (2) context of feedback, (3) learning from feedback and (4) EP-specific feedback.

#### *Nature of individualised feedback data*
The most consistent theme among participants was the need for timely feedback when specific errors were detected. This was especially important for significant clinical errors. Due to working patterns, the original prescriber might not be aware of their prescribing error at all. Feedback on a specific error within a week or two was required to optimise recall of the patient's clinical situation and the doctor's working context. More timely feedback using the EP system was a perceived advantage expressed by participants, compared with feedback based on paper drug charts and manual audit.

Participants wanted individualised feedback data to be specific, tailored and concise. They also focused on the learning aspect rather than the error itself and wanted feedback to be non-judgemental, balanced and motivational, especially if delivered by email.

> It would be nice to have a headline that says something like 'congratulations, you only made two errors this past month'—give some positive feedback as well as information about prescribing error (FY2, interview 2).

> Error rate sounds a bit judgey [sic] and feels like an attack, feels like you've been watched (FY1, focus group).

In terms of format and content, there was no consensus on preferences around the example reports shared. Instead, a combination of text, illustrations, graphs and case studies were generally considered to be useful.

#### Context of feedback
Although participants were generally positive about receiving feedback on EP errors, this depended on the context in which it was given and received. An important factor was the feedback provider. EP feedback in a report from an anonymised source was perceived to be potentially problematic due to workload pressures and limited time to focus on emails. Most participants had received verbal feedback from their ward pharmacist who had corrected an error and were comfortable with this if they had worked with them before. There was a degree of trust and confidence in particular with specialist clinical pharmacists (eg, renal, paediatrics) as they were perceived to understand the nuances of prescribing for their patient cohort and were able to better highlight clinically relevant errors. However, participants raised concern about the potential workload for their ward pharmacist if more regular formal feedback was to be delivered due to the time required for data collection and also being able to reciprocate with time to receive feedback.

> Sometimes pharmacists will say "oh you do know that you got this wrong" but it's done casually (FY2, interview 1).

There was consensus from participants that feedback on more significant prescribing errors (ie, those associated with potential or actual 'major harm') was very important. The value of feedback for less serious errors was less convincing as participants perceived that they would not have time to focus on all errors. When given a list of example error types, participants ranked prescribing a drug to which a patient has an allergy as the most important error to receive feedback on. Clinical contraindications, omissions and unnecessary prescribing were also prioritised for feedback. However, dosing errors and formulation errors were not.

#### *Learning from feedback*
Participants acknowledged that learning from prescribing feedback could take place at an individual level, in a group setting or even at organisational level.

There were mixed feelings about learning from data such as baseline error rates; some participants perceived this to be helpful for context, while others felt that statistics (especially in isolation) would not be as helpful for learning in comparison to narrative feedback. Similarly, benchmarking statistics with peers had a mixed response; some participants valued the quantitative aspect while others were unclear about how this would advance their learning and were afraid of being judged.

Error rates is like a per cent on your exams…It doesn't say anything, and you can't learn from it (FY1, focus group).

However, there was general agreement that learning about prescribing errors in the form of case studies was a helpful educational tool. Learning about errors made by peers in a group setting was useful, especially as foundation doctors rotated specialties. However, it was noted that reporting on real EP errors publicly needed to be done carefully and anonymously, as there was concern about having a negative impact:

…it might break the team spirit (FY1, focus group).

One participant was clear that all prescribing errors were significant for patient safety and that accepting minor errors was suggestive of professional 'laziness'. Instead, they felt that the organisation should be learning from all types of errors and cultivating good prescribing practice:

'We as a hospital are lazy… for example, in giving the indication for medications (FY2, interview 4).

The team prescribing culture was felt to be important, as junior doctors were influenced by senior decision-makers, particularly in terms of bad habits or prescribing outside the protocol. Moreover, prescribing decisions were often made by the consultant but enacted by the junior doctors. Therefore, one participant suggested providing feedback to the team in general as well as individuals.

The junior doctors are a lot influenced by the senior doctors and there is a hierarchy, sometimes I want to say something… but it's in some way difficult to correct them. But anyway, I just say it… because patient safety is the priority… It's the job of the entire team to look at the drug chart and review it (FY1, interview 4).

In addition, error rates on its own might not be helpful for comparing between specialties with different prescribing cultures, therefore reducing the validity of the data.

Graphs are a nice idea in some ways… but if you are in a specialty and you are not making many errors, it's weird to be like "oh well, I do way better than the other specialties" … how serious are these errors on the graphs? If you do badly it's embarrassing for the team, but it can be a good thing for some… maybe they are going to do more to do their best (FY2, interview 5)

### EP-specific feedback

Participants perceived that the EP system architecture was capable of providing greater prescribing support than paper prescribing and there was a potential for EP data to be used for feedback. Participants wanted to have a better understanding of the EP system to avoid repeating

errors and queried how EP error data would be accurately collected and judged. There was anxiety about being monitored and that automated EP data might not accurately reflect their abilities.

There were some examples of participants who had difficulties transitioning from paper-based prescribing. Specifically, participants found that the default settings for standard drug administration times made it more difficult to prescribe complex scheduling times, reschedule drug administration times and suspend medication doses. These EP-specific challenges exposed them to risks of making EP errors. Participants were keen to know how to use the EP system better in these situations and how the system could be adapted to be more intuitive.

Some participants also suggested that the national student prescribing safety assessment was not a good marker for preparedness and was not helpful for EP skills:

it's very different from actually using the system (FY2, interview 5).

## DISCUSSION
### Key findings

Foundation doctors in this mixed methods study reported that current feedback provision on EP errors was lacking or informal, and that existing EP training and resources were underused. This feedback gap has been previously identified[17] and persists despite quality improvement initiatives at the study organisation. However, these have been difficult to sustain due to the use of manual audit and human resources, including individual ward pharmacists[21 22] Other studies have also identified that provision of safe prescribing training and induction is variable (including e-learning) due to the lack of a standardised prescribing curriculum for postgraduate doctors.[23 24]

Survey participants believed feedback about prescribing errors to be important and indicated readiness to receive real-time, individualised EP feedback data. Participants in the interviews and focus group expanded that feedback also needed to be in manageable amounts, motivational and clearly signposting how to learn or improve. Survey participants wanted feedback and better EP training to prevent repeating errors. In addition to individualised EP error data, they were receptive to learning from general prescribing errors and aggregated EP data. However, focus group and interview participants revealed that despite this surface acknowledgement, there was a lack of consensus about how best to learn from statistical data. Potential limitations included concern about how the data would be collected, whether it reflected clinical ability and time to reflect. Concerns about being performance managed have also been highlighted in other patient safety research with junior doctors.[25]

Participants in the survey, interviews and focus group all identified ward pharmacists as a trusted source and provider of feedback. This is consistent with other studies evaluating pharmacist-led feedback interventions.[11 12 14 26]

## Interpretation

This study supports the findings of previous research that foundation doctors want feedback on prescribing errors.[17] There is hope that EP technology could facilitate feedback data in a timely, individualised and digitised format to optimise learning. However, recent initiatives to extract EP data for this purpose is still challenging.[27] In addition, provision of EP error data in isolation would not be adequate to promote reflective practice and prevent further EP errors. This is consistent with other research suggesting that email feedback alone is less effective than group discussions or directly observed prescribing encounters.[28] In our study, participants described contextualising EP errors using illustrative case studies and having the opportunity to debrief with a pharmacist as potential learning opportunities. Enabling novice prescribers to have enough time and cognitive space to both consider the feedback and formulate an action plan were identified as potential challenges. An electronic feedback tool involving individualised learning plans has been described by others.[29]

Participants did not agree on the use of statistical data as a feedback mechanism. However, a Cochrane review suggests that audit and feedback interventions in healthcare are an effective method of stimulating behaviour change.[30] Perceptual control theory also recognises the role of 'benchmarking' against a 'reference standard' as a key motivator for behaviour change.[28] This might be due to the study exploring doctors' perceptions rather than a direct evaluation of experience.

In addition, interview participants discussed 'feedback', but it was not always clear whether they were distinguishing between general prescribing feedback and provision of EP data. It is therefore possible that participants and researchers were discussing different aspects of 'feedback' but using the term interchangeably to mean different things. This has also been found in the educational literature, where there is well-recognised incongruence between learners and teachers as to the definition, quantity and quality of feedback given and received.[31]

## Strengths and limitations

The main strength of this study came from combining quantitative and qualitative data collection methods. This gave us the opportunity to explore the research question from different perspectives. The survey was completed by more participants and captured a broader range of respondents, while the interviews and focus group gave us an opportunity to probe responses in more depth. Eliciting the challenges perceived by foundation doctors was a strength of the qualitative data, as there were some underlying anxieties that were not readily captured by the survey data. The multidisciplinary nature of the research team (doctor and pharmacists, with medical education and medication safety expertise) was also a strength as we were able to contribute different perspectives to the data analysis from the context of our professional experience.

Limitations of this study include the low response rate and small sample size from a single organisation, which limits generalisability. It was not feasible to expand recruitment to other institutions or extend the timeline of the study. However, themes were being repeated, suggesting that the data was approaching saturation. There are differences between FY1 (newly qualified) and FY2 (independently registered) doctors, so including both cohorts in this study population may make it more difficult to design tailored interventions. The questionnaire was not psychometrically tested. There may be participation bias, particularly with the interviews and focus group, as respondents may be more likely to be interested in the study concept. They may also have provided socially desirable responses rather than their true viewpoints. We explored participants' perceptions rather than conducting a direct evaluation in practice. Participants therefore made hypothetical assumptions about the capability of the EP system and provision of EP data. The examples that participants were shown in the interviews were based on the existing literature at the time, and thus were mainly paper-based.

## Implications for practice

This data informs future work for developing individualised, digital EP feedback data for prescribers. Feedback needs to be timely, concise and relevant, with a blended educational approach in terms of providing data and example case studies, both verbally and electronically. Situational factors should be acknowledged, and feedback should be supportive and non-judgemental. Consideration should be given to the role of pharmacists in providing feedback, allocating sufficient time for prescribers to focus on feedback, prioritising the relative importance of EP errors and support formulation of action plans. Other learning points include improving awareness and engagement of EP training opportunities, changing the focus to prescribing accuracy (rather than on errors), maximising opportunity for reflective practice (not just providing information) and enabling individuals to contribute to organisational learning and improvements (eg, reporting EP system defaults). A challenge to achieving this in practice includes staffing and time resource, as well as digital capability.

## Future research

Future research should focus on the evaluation of EP feedback mechanisms and integrating this into clinical practice for novice prescribers. The feasibility, acceptability and use of these tools should be considered and appropriate mechanisms for providing context and promoting learning put into place. Challenges, unintended consequences and resource implications of introducing a novel EP feedback interface should also be captured. Our study focused on junior doctors as the main prescribers in hospitals, but in the UK, independent nurses and pharmacist prescribers also have an important role, thus provision of prescribing feedback to multiprofessional prescribers should be explored. Future educational interventions

could consider combining individualised EP feedback data with modern initiatives including simulation and video-stimulated feedback.[13 32]

## CONCLUSION

The digital transformation of hospital inpatient prescribing presents a timely opportunity to develop, implement and evaluate an individualised prescribing feedback and learning initiative to reduce prescribing errors. Foundation doctors in this study are receptive to receiving value feedback on their prescribing, keen to learn from errors to develop their clinical prescribing skills and want to use the electronic interface effectively.

Our findings suggest a range of educational options would be a positive strategy to enhance the value of individualised EP feedback data for novice prescribers. We found a demand for EP technology to enable provision of real-time feedback to support learning, thus potentially reducing prescribing errors. The next steps are to develop a feasible approach for extracting EP data, providing individualised EP feedback data and evaluating its use and effectiveness in practice.

**Contributors** MM and BDF conceived the study and study design. Methods were developed by MM, BDF, GD and AK. Data were collected by AK, GD and MM and initial data analyses were conducted by GD, AK, AC and MM. Interpretation of the data was discussed to consensus by AC, BDF and MM. The manuscript was prepared and revised by AC and MM. All authors reviewed and finalised the manuscript. AC is the author acting as guarantor.

**Funding** This study was supported by the National Institute for Health Research (NIHR) Imperial Patient Safety Translational Research Centre (PSTRC-2016-004). AC is an NIHR-funded Academic Clinical Fellow. The views expressed are those of the author(s) and not necessarily those of the NIHR or the Department of Health and Social Care.

**Competing interests** BDF supervised a PhD student who was part funded by Cerner (a supplier of electronic health record systems) but whose work was unrelated to this study.

**Patient and public involvement** Patients and/or the public were not involved in the design, or conduct, or reporting, or dissemination plans of this research.

**Patient consent for publication** Not applicable.

**Ethics approval** This study met the Health Research Authority criteria for service evaluation; National Health Service ethics approval was not required. It was registered as a service evaluation within the trust (reference 266). The study was conducted in line with ethical research practice: ensuring voluntary participation, obtaining written consent for interviews and focus groups and anonymising data in line with data protection regulations. There was no patient and public involvement in this research. Participants gave informed consent to participate in the study before taking part.

**Provenance and peer review** Not commissioned; externally peer reviewed.

**Data availability statement** Data are available upon reasonable request.

**ORCID iDs**
Ann Chu http://orcid.org/0000-0002-8894-4238
Monsey McLeod http://orcid.org/0000-0001-5571-2417

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
