## [Reviewer comments · BMJ Open]

ARTICLE DETAILS

TITLE (PROVISIONAL)	Learning from electronic prescribing errors: a mixed methods study of junior doctors' perceptions of training and individualised feedback data
AUTHORS	Chu, Ann; Kumar, Arika; Depoorter, Geraldine; Franklin, Bryony; McLeod, Monsey

VERSION 1 – REVIEW

REVIEWER	Kevin Fuji Creighton University
REVIEW RETURNED	24-Oct-2021

GENERAL COMMENTS	This manuscript describes a mixed methods study exploring the attitudes of junior physicians toward feedback about electronic prescribing (e-prescribing) errors. The manuscript is well-written and the topic continues to be important. E-prescribing errors continue to persist, and as the authors state, much of the focus thus far has been technology-oriented with less focus on the actual end-users of the system. There are numerous weaknesses to the manuscript, including potentially major ones related to the methodology as noted below. Title 1) It appears that personalized feedback was a finding of the study, rather than being an explicit a priori focus (which would be aligned with the qualitative phase of the study). If this is correct, the word “personalised” should be removed from the title. Methods 2) Setting: Information should be provided about if any of the EP training noted is required for junior physicians. 3) Participants: Breakdown should be provided of how many physicians were in FY1 vs. FY2. 4) Recruitment: All of this information would be better integrated into the Data Collection sub-section. 5) Recruitment, Paragraph 2: The information in this paragraph raises questions about the methods used. A justification should be provided for the use of interviews or focus groups beyond participant preference. These are two different types of qualitative data collection methods that are chosen for different purposes and can yield different results. They are not always interchangeable as focus groups support more interaction and benefits from conversations between participants. This is in contrast to interviews which are done one-on-one and support more privacy (potentially important in this
--

	study, especially given the limitation of social desirability noted). Additionally, were Phase 2 participants a sub-set of the Phase 1 participants? This is what is typically done in a sequential explanatory design. If this is not what was done, more information should be provided included a justification for why an “atypical” sampling for Phase 2 was used. 6) Data Collection, Phase 1: Provide details about the already-existing questionnaire used in Phase 1. What was the content that needed to be adapted to EP? Was the original questionnaire psychometrically tested? Essentially, why was this an appropriate instrument to utilize? How were the clinical scenarios developed? Who participated in their development? Were they pilot tested? Was there any face validity or content validity steps taken? 7) Data Collection, Phase 2: How specifically was the topic guide informed by the findings from Phase 1? 8) Data Analysis, Phase 1: What was the justification for combining the positive and negative responses and disregarding the neutral responses? 9) Data Analysis, Phase 2: Significantly more detail needs to be provided for the qualitative analysis. For example, how many researchers coded the data? What type of coding framework was employed? How were codes grouped to form themes? Were any qualitative validation measures (e.g. member checking) used? Was data saturation or thematic saturation achieved? For the mixed analysis, what does it mean to use a “triangulation approach”? Don’t assume that all readers are familiar with mixed methods. All data analysis phases of the study need to be described rigorously to ensure that each phase was methodologically sound. Results 10) Phase 2, Nature of personalised feedback: The description of this theme focuses much more on timely feedback compared to personalised feedback. Authors may want to reconsider the name of this theme. Additionally, it is noted that “More timely feedback using the EP system was seen as an advantage...” However, is that technically feasible for the EP system? 11) EP-specific feedback: What about the default settings for standard drug administration times made it more difficult to prescribe accurately? Discussion 12) Overall, the authors seem to have missed opportunities to connect the findings to the wider literature base. For example, one of the potential strategies discussed was peer comparison, and there is a large body of literature noting the impact of that approach for physician behavioral change. Interprofessional collaboration between physicians and pharmacists to support patient safety is also described in the existing literature, and this was an emphasized finding of the current study. The Discussion should also consider commenting on enhancing required and optional training resources.
--	---

REVIEWER	Andrea Huseh-Zosel North Dakota State University Nursing Program, Public Health
REVIEW RETURNED	26-Apr-2022

GENERAL COMMENTS	This paper fills a gap in the literature related to electronic prescribing training, and extends lessons learned from previous studies on prescribing errors focused on paper-based prescribing. Background: This section would benefit from the inclusion of more recent references (e.g. McLeod, M., Farah, S., Macaulay, K., Sheth, T., Patel, M., Ghafour, A., ... & Franklin, B. (2021). Designing a continuous data-driven feedback and learning initiative to improve electronic prescribing: an interdisciplinary quality improvement study. International Journal of Pharmacy Practice, 29(Supplement_1), i18-i19. Lloyd, M., Watmough, S. D., O'Brien, S. V., Hardy, K., & Furlong, N. (2021). Evaluating the impact of a pharmacist-led prescribing feedback intervention on prescribing errors in a hospital setting. Research in Social and Administrative Pharmacy, 17(9), 1579-1587.) Methods: Please list the number of focus groups conducted. Results: For Phase 2 results, providing a table with additional quotes from study participants within the four identified themes would provide readers further context related to responses provided within the interviews/focus groups. Abstract/Results: The abstract lists 5 focus group participants and 7 interview participants, but the results state 7 focus group participants and 5 interview participants. Clarify and correct respective sections. Discussion: Future research could examine differences in EP training/feedback preferences by foundation year. You had mentioned on page 12, line 3 that FY1 doctors may not have had "direct experience of prescribing in any other setting." This might also be included as a limitation of the study.
--

REVIEWER	Mari Kannan Maharajan International Medical University, Pharmacy
REVIEW RETURNED	27-Apr-2022

GENERAL COMMENTS	Overall, the manuscript focused on the junior doctors and their training and feedback related to EP. The study design is appropriate and the data collected were interpreted correctly. It would be better if the introduction section focused more on the need of the studies based on the latest literature. There are recent research articles on patient safety and prescribing errors to refer. Also, a clear justification for using Explanatory sequential mixed method for this research need to be included. The study selected only three acute hospitals. Is there any specific reasons for the selection? Does the sample size involved in the study is sufficient to generalise the data? Please explain. The authors mentioned about this in the limitation. However, a better justification to explain why the study used only a smaller population may add more value to the
--

	manuscript. According to the results, foundation doctor reported that current feedback provision on EP errors was lacking or informal. Any latest literature support this claim? Authors are encouraged to discuss the potential reasons for the existing EP training and resources were underutilised in detail. Authors can provide few possible solutions to address user-centred gaps to support junior doctors' learning.
--	---

VERSION 1 – AUTHOR RESPONSE

Reviewer 1 comments	Authors' response
Title 1) It appears that personalized feedback was a finding of the study, rather than being an explicit a priori focus (which would be aligned with the qualitative phase of the study). If this is correct, the word “personalised” should be removed from the title.	Thank you for this feedback. As stated in our objectives, we did intend to research doctors' readiness to receive personalised feedback about their own EP errors. This was explored in both phases 1 and 2. However, we have amended this to 'individualised feedback data' throughout the text to make this clearer.
Methods 2) Setting: Information should be provided about if any of the EP training noted is required for junior physicians.	This is a good point and we have amended the text to make this clearer.
3) Participants: Breakdown should be provided of how many physicians were in FY1 vs. FY2.	Participant invitations were circulated by the Trust postgraduate education team and blinded to the research team. The year groups are approximately equal in size.
4) Recruitment: All of this information would be better integrated into the Data Collection subsection.	This has been amended as suggested.
5) Recruitment, Paragraph 2: The information in this paragraph raises questions about the methods used. A justification should be provided for the use of interviews or focus groups beyond participant preference. These are two different types of qualitative data collection methods that are chosen for different purposes and can yield different results. They are not always interchangeable as focus groups support more interaction and benefits from conversations between participants. This is in contrast to interviews which are done one-on-one and support more privacy (potentially important in this study, especially given the limitation of social	The reviewer raises a good point and we have now amended the text to clarify our justification for including both methods. This was about maximising recruitment and generating insights from participant interactions in the focus group, in addition to in-depth individual views in interviews. Comparison between the data sets suggested that themes were similar and could be integrated.

desirability noted).	
Additionally, were Phase 2 participants a sub-set of the Phase 1 participants? This is what is typically done in a sequential explanatory design. If this is not what was done, more information should be provided included a justification for why an “atypical” sampling for Phase 2 was used.	In ‘study design’, we explained recruitment for both phase 1 and 2 were from the same population. We have amended the text to clarify this in ‘data collection’ also. Due to the anonymous nature of the survey, we did not confirm if any of the participants contributed to both phases. However, we decided that maximising recruitment for phase 2 was essential and thus invited the whole population to participate.
6) Data Collection, Phase 1: Provide details about the already-existing questionnaire used in Phase 1. What was the content that needed to be adapted to EP? Was the original questionnaire psychometrically tested? Essentially, why was this an appropriate instrument to utilize?	The published questionnaire is referenced. We have amended the text to clarify that changes were made to reference EP from paper-based prescribing. The questionnaire was not psychometrically tested. This has been added to the limitations section.
How were the clinical scenarios developed? Who participated in their development? Were they pilot tested? Was there any face validity or content validity steps taken?	Thank you for this feedback. We have amended the text to clarify that the scenarios were developed by research pharmacists. They were pilot tested by four pharmacists and one doctor for face and content validity.
7) Data Collection, Phase 2: How specifically was the topic guide informed by the findings from Phase 1?	The text has been modified to clarify that the topic guide was formulated after an interim analysis of survey data in phase 1 and areas of further interest identified.
8) Data Analysis, Phase 1: What was the justification for combining the positive and negative responses and disregarding the neutral responses?	The text has been amended to clarify that neutral responses were still included in the figures.
9) Data Analysis, Phase 2: Significantly more detail needs to be provided for the qualitative analysis. For example, how many researchers coded the data? What type of coding framework was employed? How were codes grouped to form themes? Were any qualitative validation measures (e.g. member checking) used? Was data saturation or thematic saturation achieved? For the mixed analysis, what does it mean to use a “triangulation approach”? Don’t assume that all readers are familiar with mixed methods. All data analysis phases of the study need to be described rigorously to ensure that each phase was methodologically sound.	Thank you for this feedback. The text has been significantly modified to provide more detail. Phase 2 interview and focus group notes were transcribed and then analysed thematically using an inductive approach. This involved systematic coding of the transcripts, organising codes into categories followed by grouping into broader themes. Two researchers conducted this independently, with quality assurance by a third, more experienced researcher. Comparison between the interview and focus group data sets suggested that themes were similar and could be integrated. Themes were discussed to consensus. Member checking was not used.

	The quantitative and qualitative data were then further integrated at the discussion phase. This included reflection on whether the data was complimentary or if there were areas of divergence. This was discussed to consensus by the wider research team. The term 'triangulation approach' has been removed.
Results 10) Phase 2, Nature of personalised feedback: The description of this theme focuses much more on timely feedback compared to personalised feedback. Authors may want to reconsider the name of this theme. Additionally, it is noted that "More timely feedback using the EP system was seen as an advantage..." However, is that technically feasible for the EP system?	As in point 1, 'personalised' has been changed to 'individualised feedback data' throughout the text. The text has been amended to make clearer that the 'timely feedback using the EP system' was a perceived advantage from the study participants.
11) EP-specific feedback: What about the default settings for standard drug administration times made it more difficult to prescribe accurately?	The paragraph has been re-organised to make this clearer. Specifically, participants found that the default settings for standard drug administration times made it more difficult to prescribe complex scheduling times, reschedule drug administration times, and suspend medication doses. These EP-specific challenges exposed them to risks of making EP errors.
Discussion 12) Overall, the authors seem to have missed opportunities to connect the findings to the wider literature base. For example, one of the potential strategies discussed was peer comparison, and there is a large body of literature noting the impact of that approach for physician behavioral change. Interprofessional collaboration between physicians and pharmacists to support patient safety is also described in the existing literature, and this was an emphasized finding of the current study. The Discussion should also consider commenting on enhancing required and optional training resources.	The discussion has been extensively modified to address the reviewer's comments, including provision of training resources, the role of audit and the role of pharmacists. Updated literature has been included and connected to the study results.
Reviewer 2 comments	

Background: This section would benefit from the inclusion of more recent references (e.g. McLeod, M., Farah, S., Macaulay, K., Sheth, T., Patel, M., Ghafour, A., ... & Franklin, B. (2021). Designing a continuous data-driven feedback and learning initiative to improve electronic prescribing: an interdisciplinary quality improvement study. International Journal of Pharmacy Practice, 29(Supplement_1), i18-i19. Lloyd, M., Watmough, S. D., O'Brien, S. V., Hardy, K., & Furlong, N. (2021). Evaluating the impact of a pharmacist-led prescribing feedback intervention on prescribing errors in a hospital setting. Research in Social and Administrative Pharmacy, 17(9), 1579-1587.)	Thank you for this feedback. The introduction has been restructured to signpost the literature gap in relation to EP feedback for junior doctors more clearly. Updated references have been included including the ones highlighted here. McLeod 2021 has been cited in the discussion.
Methods: Please list the number of focus groups conducted.	The text has been amended to clarify that one focus group was conducted.
Results: For Phase 2 results, providing a table with additional quotes from study participants within the four identified themes would provide readers further context related to responses provided within the interviews/focus groups.	A table with the thematic framework has now been added to the appendix (due to limitation on figures and tables in the main paper).
Abstract/Results: The abstract lists 5 focus group participants and 7 interview participants, but the results state 7 focus group participants and 5 interview participants. Clarify and correct respective sections.	Thank you for spotting this error. The abstract has been corrected.
Discussion: Future research could examine differences in EP training/feedback preferences by foundation year. You had mentioned on page 12, line 3 that FY1 doctors may not have had “direct experience of prescribing in any other setting.” This might also be included as a limitation of the study.	This is now acknowledged in limitations.
Reviewer 3 comments	

It would be better if the introduction section focused more on the need of the studies based on the latest literature. There are recent research articles on patient safety and prescribing errors to refer.	Thank you for this feedback. The introduction has been restructured to signpost the literature gap in relation to EP feedback for junior doctors more clearly. Updated references have been included.
Also, a clear justification for using Explanatory sequential mixed method for this research need to be included.	This is now clarified under 'study design'. This study design was chosen to permit analysis of the quantitative data for broad perspectives followed by in-depth exploration of themes that required further explanation.
The study selected only three acute hospitals. Is there any specific reasons for the selection?	The text has been amended to clarify that this study focussed on 3 hospitals belonging to one organisation so that participants were referring to the same EP system and had equal availability of training resources.
Does the sample size involved in the study is sufficient to generalise the data? Please explain. The authors mentioned about this in the limitation. However, a better justification to explain why the study used only a smaller population may add more value to the manuscript.	The text has been expanded to explain further the limitations. It was not feasible to expand recruitment to other organisations or extend the timeline of the study. Themes were starting to be repeated which suggested thematic saturation.
According to the results, foundation doctor reported that current feedback provision on EP errors was lacking or informal. Any latest literature support this claim? Authors are encouraged to discuss the potential reasons for the existing EP training and resources were underutilised in detail.	The discussion has been modified to address this with reference to updated literature.
Authors can provide few possible solutions to address user-centred gaps to support junior doctors' learning.	This is discussed under the section 'Implications for Practice'.

VERSION 2 – REVIEW

REVIEWER	David Cunningham National Health Service Education for Scotland Knowledge Network, Medicine Directorate
REVIEW RETURNED	09-Oct-2022
GENERAL COMMENTS	The authors have considerable modifications to the original manuscript. I think it is improved considerably and should now be published.